# Lack of RAC1 in macrophages protects against atherosclerosis

Sashidar Bandaru[1☯], Chandu Ala[1☯], Matias Ekstrand[2], Murali K. Akula[2,3], Matteo Pedrelli[4], Xi Liu[1], Göran Bergström[2,5], Liliana Håversen[2], Jan Borén[2], Martin O. Bergo[3,6], Levent M. Akyürek[1,7]*

1 Department of Laboratory Medicine, Institute of Biomedicine, Sahlgrenska Academy at University of Gothenburg, Gothenburg, Sweden, 2 Department of Molecular and Clinical Medicine, Institute of Medicine, Sahlgrenska Academy at University of Gothenburg, Gothenburg, Sweden, 3 Sahlgrenska Cancer Center, Sahlgrenska Academy at University of Gothenburg, Gothenburg, Sweden, 4 Division of Clinical Chemistry, Department of Laboratory Medicine, Karolinska Institute, Stockholm, Sweden, 5 Department of Clinical Physiology, Västra Götalandregionen, Sahlgrenska University Hospital, Gothenburg, Sweden, 6 Department of Biosciences and Nutrition, Karolinska Institute, Stockholm, Sweden, 7 Department of Clinical Pathology, Västra Götalandregionen, Sahlgrenska University Hospital, Gothenburg, Sweden

☯ These authors contributed equally to this work.
* levent.akyurek@gu.se

**Data Availability Statement:** All relevant data are within the manuscript and its Supporting Information files.

**Funding:** This work was supported by the Strategic Fund from the Institute of Biomedicine, University

## Abstract

The Rho GTPase RAC1 is an important regulator of cytoskeletal dynamics, but the role of macrophage-specific RAC1 has not been explored during atherogenesis. We analyzed RAC1 expression in human carotid atherosclerotic plaques using immunofluorescence and found higher macrophage RAC1 expression in advanced plaques compared with intermediate human atherosclerotic plaques. We then produced mice with *Rac1*-deficient macrophages by breeding conditional floxed *Rac1* mice (*Rac1^fl/fl^*) with mice expressing Cre from the macrophage-specific lysosome M promoter (*LC*). Atherosclerosis was studied *in vivo* by infecting *Rac1^fl/fl^* and *Rac1^fl/fl^*/*LC* mice with AdPCSK9 (adenoviral vector overexpressing proprotein convertase subtilisin/kexin type 9). *Rac1^fl/fl^*/LC macrophages secreted lower levels of IL-6 and TNF-α and exhibited reduced foam cell formation and lipid uptake. The deficiency of *Rac1* in macrophages reduced the size of aortic atherosclerotic plaques in AdPCSK9-infected *Rac1^fl/fl^*/*LC* mice. Compare with controls, intima/media ratios, the size of necrotic cores, and numbers of CD68-positive macrophages in atherosclerotic plaques were reduced in *Rac1*-deficient mice. Moreover, we found that RAC1 interacts with actin-binding filamin A. Macrophages expressed increased RAC1 levels in advanced human atherosclerosis. Genetic inactivation of RAC1 impaired macrophage function and reduced atherosclerosis in mice, suggesting that drugs targeting RAC1 may be useful in the treatment of atherosclerosis.

## Introduction

Atherosclerosis is a slowly progressive chronic inflammatory disorder which develops in response to hyperlipidemia [1]. Although immune cells, particularly bone marrow-derived

of Gothenburg, the Swedish Cancer Foundation (Contract number 17 0171), and the ALF fund (ALFGBG-495961) from the Sahlgrenska Academy Hospital in the Västra Götalandregionen to LMA.

**Competing interests:** The authors have declared that no competing interests exist.

**Abbreviations:** AdPCSK9, adenoviral vector overexpressing proprotein convertase subtilisin/ kexin type 9; BMMs, bone marrow macrophages; FLNA, filamin A; FLNA^CT, C-terminal fragment of FLNA; LC, lysosome M promoter; mmLDL, minimally oxidized-LDL; oxLDL, oxidized LDL; RAC1, RAS-related C3 botulinum toxin substrate 1; Rac1^fl/fl, mice homozygously expressing the floxed *Rac1* gene; *Rac1^fl/fl*/LC, mice lacking *Rac1* in macrophages; SMCs, smooth muscle cells.

macrophages (BMMs), contribute to the development of atherosclerotic plaque, the underlying molecular mechanisms are not fully explored. Cholesterol deposition into the arterial wall triggers BMMs to engulf the lipids to form foam cells [2]. The formation of foam cells and thereby buildup of vascular plaques gradually results in either plaque rupture or narrowing of the blood vessels, subsequently blocking oxygen supply to the heart as well as other vital organs that in turn lead to severe complications such as stroke and cardiac and renal failure [3]. Thus, studying macrophage involvement in atherosclerosis with genetically engineered mouse models may give us a better understanding of macrophage function, thereby providing new insights to develop new clinical targets to stop or at least slow atherosclerotic development.

RAS-related C3 botulinum toxin substrate 1 (RAC1) is a member of the RAC family of GTPases, a subfamily of RHO proteins. Among the three isoforms, RAC1 is ubiquitously expressed and extensively studied due to the involvement in regulating cell migration, phagocytosis, inflammatory responses, actin remodeling and gene expression [4]. Missense mutations in the human *RAC1* gene are accompanied with cardiac abnormalities [5]. Complete inhibition of *Rac1* in mice results in embryonic lethality at a very early stage due to defects in germ layer formation, indicating that RAC1 expression is critical for organogenesis [6]. Cardiomyocyte-specific deletion of *Rac1* decreases myocardial hypertrophy by reducing oxidative stress through NADPH inhibition [7]. Recent observations indicate that RAC1 could be a potential target in atherosclerosis. Firstly, RAC1 induces systemic inflammation by interacting with various proteins and inhibition of RAC1 reduces inflammatory response which could reduce the burden of atherosclerotic plaques [8]. Secondly, RAC1 induces actin reorganization during cell membrane ruffling and phagocytosis [9] and inhibition of targeting actin dynamics in BMMs could be a potential target for treating atherosclerosis [10]. Finally, actin remodeling partly regulates lipid uptake in macrophages, RAC1 targets actin-binding protein filamin A (FLNA) to remodel the cytoskeleton [11] and FLNA is involved in both of these processes [12].

Elevated RAC1 expression in intimal macrophages from human carotid endarterectomies with advanced atherosclerotic lesions prompted us to hypothesize that inhibiting RAC1 could reduce atherosclerosis *in vivo*. To do this, we generated mice deficient for RAC1 in macrophages using the *Cre-loxP* system and induced atherosclerosis to define the consequences of *Rac1*-deficiency on macrophage function and plaque development. We have further provided new insights into the role of RAC1 in macrophages.

## Materials and methods

### Ethics statement

The animal experiments were conducted in accordance with the guidelines of the Animal Care and Use Committee of the University of Gothenburg. This investigation was approved by Institutional Review Board of Swedish Board of Agriculture and conforms to the research animal Directive 2019/2649 for mouse breeding and 2016/59 for mouse model of atherosclerosis. All mice were euthanized by placing them into an induction chamber (Abbott Scandinavia AB) up to 10 min, where mice inhaled 4% isoflurane (Forene isoflurane, Abbott Scandinavia AB) and killed by cervical dislocation and surgical excision of the heart. These animals were housed two to three per cage with free access to food and water on a 12 h light/dark cycle. Every effort was made to minimize the number and suffering of animals.

### Human carotid arteries

Human intermediate (type III) and advanced (type VI) atherosclerotic plaques (obtained from the same human carotid endarterectomies as previously described [13]) were stained using

multiple immunofluorescence with the anti-Mac-2 antibody to detect macrophages (Cedarlane), as well as, RAC1 (Sigma-Aldrich) and SM22α antibodies (Abcam) to detect smooth muscle cells (SMCs) (n = 9 in each group). Alexa 488 anti-mouse (Jackson ImmunoResearch), Alexa 594 anti-rat (Jackson ImmunoResearch), and Alexa 647 anti-goat (Jackson ImmunoResearch) were applied as secondary antibodies. Nuclear staining was done by DAPI. the JACoP Plugin [14] using ImageJ Software [15] was used to analyze overlapping RAC1 signals with macrophages or SMCs in the intimal area of both intermediate and advanced plaques. The same signal intensity threshold was applied to all analyzed images to measure RAC1 levels in macrophages and SMCs between the different groups.

## Mice

All mouse experimental procedures were followed in accordance with institutional guidelines of University of Gothenburg, Sweden. To produce mice that are deficient for RAC1 in macrophages, female C57BL/6 mice homozygously expressing the floxed *Rac1* gene (*Rac1*$^{fl/fl}$) [6] were crossbred with male mice homozygously expressing Cre under the monocyte-specific lysozyme M promoter (*LC*) [12]. Mice were genotyped for *Rac1* deficiency as previously described [16].

## Primary murine bone marrow-derived macrophages

Bone marrow-derived monocytes were extracted from both femur and tibia of the mice as described elsewhere [10]. To differentiate the monocytes into macrophages, cells were cultured with DMEM medium containing 10% FBS, 1% glutamine, 1% gentamicin, 1% HEPES, 0.01% β-mercaptoethanol, and 10% CMG14-12 cell line supernatant as a source of mouse M-CSF. One week later, BMMs were plated and assayed at different time points. BMMs were isolated from *Rac1* (*Rac1*$^{fl/fl}$) or lacking *Rac1* (*Rac1*$^{fl/fl}$/LC) mice as described elsewhere [10]. To analyze the purity of the isolated BMMs, *Rac1* (F 5′ –AGAGTACATCCCCACCGTCTT–3′ and R 5′ – GTCTTGAGTCCTCGCTGTGT–3′ ), *CD68* (F 5′ –ACCTACATCAGAGCCCGAGT–3′ and R 5′ –CGCCTAGTCCAAGGTCCAAG–3′ ), and *SM22α* [17] expression were assessed using total RNA extracted from BMMs and reverse transcription polymerase chain reaction (RT-PCR). 18S transcripts were included as internal loading controls. Purity of BMMs was shown by CD68 immunofluorescence staining as previously described [12]. BMMs were then stained by immunofluorescence using an anti-actin phalloidin antibody (Life Technologies) as well as FLNA$^{CT}$ (Bethyl Laboratories). To measure cell elongation ratio, images of *Rac1*$^{fl/fl}$ and *Rac1*$^{fl/fl}$/LC BMMs were captured by an Axio Imager M1 microscope (Zeiss) and cell length and width were measured using Biopix software (GU Ventures). BMMs were stimulated with 10 ng/ml LPS 15 minutes before immunoblotting and 8 hours before ELISA. Macrophage cell lysates were immunoblotted with primary antibodies directed against RAC1 (Merck Millipore), FLNA (Chemicon, Bethyl Laboratories), SR-B1 (Santa Cruz), LXRα/β, CD36, COX2, ABCG1, and ABCA1 (Novus Biologicals) [18]. The band densities were quantified by Image-Quant software (Bio-Rad) in at least triplicated experiments. For co-immunoprecipitation assay, total cell lysates were isolated from BMMs, immunoprecipitated with either FLNA$^{CT}$ antibody (Novus Biologicals) or RAC1 and then immunoblotted against RAC1 or FLNA$^{CT}$ according to the manufacturer´s protocol (Thermo Scientific). Secreted levels of IL-6, IL-10, IL-12 and TNF-α were detected by ELISA (eBioscience) from either mouse serum or cultured BMMs treated by LPS [10]. Using modified Boyden chambers, proliferation and migration of BMMs were assessed up to 6 days and 18 hours, respectively [12].

## Mouse model of atherosclerosis by adenoviral infection with AdPCSK9

To induce atherogenesis by hypercholesterolemia, 8-10-week-old male *Rac1*<sup>fl/fl</sup> or *Rac1*<sup>fl/fl</sup>/*LC* mice were infected with adenoviral vector overexpressing proprotein convertase subtilisin/kexin type 9 (AdPCSK9) at a dose of $2 \times 10^{11}$ viral particles/mouse [19]. These mice were then fed a high-fat diet for 20 weeks and whole aortas were then fixed, pinned and stained with Sudan IV [20]. Paraffin-embedded mouse aortic arches were sectioned and stained using immunofluorescence with antibodies against CD68 (Abcam) and SM22α (Abcam) and secondary antibodies rat (Jackson ImmunoResearch), and Alexa 647 anti-goat (Jackson ImmunoResearch) [13]. Immunofluorescence staining intensity was quantified in the entire wall of aortic arches as reported elsewhere [12]. Pinned whole aortas and aortic arch histological images were captured with a Leica Microsystems microscope and plaques with lipid content was quantified using Biopix software (GU Ventures) [10]. In Hematoxylin & Eosin (H&E) stained cross sections of aortic arches, intima/media ratios were quantified as described earlier [12]. Sirius red stain was performed as described earlier [19]. The percentage of Sirius red positivity and necrotic core areas within intimal thickening formed in *Rac1*<sup>fl/fl</sup> and *Rac1*<sup>fl/fl</sup>/*LC* aortas were quantified using Biopix software (GU Ventures).

## Analysis of plasma lipid lipoproteins

Lipoproteins were separated from 2.5 μL of individual plasma samples by size-exclusion chromatography using a Superose 6 PC 3.2/30 column (GE Healthcare BioSciences AB). The sephadex column separated the lipoproteins into fractions of very low-density lipoproteins (VLDL), low-density lipoproteins (LDL), and high-density lipoproteins (HDL). Triglycerides (TG) and total cholesterol concentrations were calculated after integration of the individual chromatograms [21, 22], generated by the enzymatic-colorimetric reaction (Cholesterol CHOD-PAP and TG GPO-PAP kits, Roche Diagnostics).

## Foam cell formation assay

For foam cell formation, BMMs were incubated with 50 μg/mL minimally oxidized-LDL (mmLDL, Kalen Biomedical) for 24 hours and accumulated intracellular mmLDL were detected by Oil-Red-O staining. Images were captured by a Zeiss light microscope and analyzed using Biopix software (GU Ventures) [10].

## Lipoprotein uptake assay

LDL was prepared as previously described [23]. Oxidized LDL (oxLDL) was prepared by incubating LDL with 5 μM $CuSO_4$ at 37°C for 8 and 24 h. LDL and oxLDL were labeled with dil and BMMs were treated for 3 h with 10 μg/ml dil-labeled LDL or dil-oxLDL as previously described [23], with the exception that incubation was done in the absence of oleic acid.

## Confocal microscopy

BMMs were cultured in 4-well chamber slides and double immunofluorescence performed with RAC1 and FLNA antibodies to determine co-localization, as previously described [13].

## Statistics

To determine the difference in cell-specific RAC1 expression levels in intermediate and advanced human carotid artery plaques, Mander's overlap coefficients were calculated for each sample as described earlier [14]. Due to the paired nature of the samples, a Wilcoxon signed rank test was used to assess statistical significance.

Comparisons between multiple groups were evaluated using ANOVA and the Tukey-Kramer modification of Tukey's test, and comparisons between two groups were evaluated with Student's *t* test using GraphPad Prism 7.02 (GraphPad Software). All results were reported as means ± SEM. A *P* value ≤0.05 was considered to be statistically significant.

## Results

### Expression of RAC1 is increased in the intimal macrophages of advanced human atherosclerotic plaques

We observed that RAC1 was expressed within the intimal thickening of both intermediate and advanced atherosclerotic plaques of human carotid endarterectomies (**Fig 1A**). Co-localization studies indicated that RAC1 was expressed by intimal macrophages in atherosclerotic plaques (**Fig 1A**, right lower panel inset). An increased number of intimal macrophages were detected in advanced atherosclerotic lesions compared to intermediate plaques (100% *versus* 517%, p<0.05, **Fig 1B**). RAC1 expression was increased in intimal macrophages within the advanced atherosclerotic lesions compared to the macrophages within the intermediate lesions (0.41 *versus* 0.37 Mander´s overlap coefficient, p<0.05, **Fig 1C**). In addition to macrophages, RAC1 was expressed in vascular SMCs. Although not statistically significant, fewer RAC1-expressing SMCs were found in advanced lesions compared to intermediate lesions (0.75 *versus* 0.68 Mander´s overlap coefficient, p = 0.20, **S1 Fig in S1 File**).

### Deficiency of RAC1 alters macrophage cell shape

We produced mice with *Rac1*-deficient macrophages (*Rac1$^{fl/fl}$/LC*) and extracted BMMs either from control *Rac1$^{fl/fl}$* or *Rac1$^{fl/fl}$/LC* mice. *Rac1$^{fl/fl}$/LC* mice were fertile and did not show any pathology. Neither *Rac1* mRNA (**Fig 2A**) nor RAC1 protein (**Fig 2B**) were detected in *Rac1$^{fl/fl}$/LC* BMMs. These macrophages expressed *CD68* mRNA and lacked *SM22α* mRNA, indicating the purity of the cultured BMMs (**Fig 2A**). Compared to *Rac1$^{fl/fl}$* control BMMs, *Rac1$^{fl/fl}$/LC* BMMs were more elongated as detected by immunofluorescence staining of actin phalloidin (**Fig 2C**, left panel) and quantification of cell length and width (**Fig 2C**, right graphs) (3.28 ± 0.06 folds *versus* 2.58 ± 0.04 folds, p<0.01). There was no difference in proliferation rates between *Rac1$^{fl/fl}$* or *Rac1$^{fl/fl}$/LC* BMMs, as assayed up to 5 days (**S2A Fig in S1 File**). Stimulation with LPS did not significantly alter the number of migrated *Rac1$^{fl/fl}$* or *Rac1$^{fl/fl}$/LC* BMMs after 4 and 16 hours (**S2B Fig in S1 File**).

### Mice lacking RAC1 in macrophages develop smaller aortic atherosclerotic plaques

We observed a 33% reduction in the size of atherosclerotic plaques in whole aortas of *Rac1$^{fl/fl}$/LC* mice infected with AdPCSK9 compared with *Rac1$^{fl/fl}$* control mice (p<0.01, **Fig 3A**). Similarly, a 31% reduction in ratio of intima/media was observed in *Rac1$^{fl/fl}$/LC* aortic arches (p<0.05, **Fig 3B**). Compared to *Rac1$^{fl/fl}$* aortic controls, necrotic core areas formed within intimal thickening were reduced in *Rac1$^{fl/fl}$/LC* aortas by 81% (p<0.05, **Fig 3C**). In these aortic arches, the number of CD68-positive macrophages was reduced by 30.2% compared with *Rac1$^{fl/fl}$* control aortic arches (p = 0.05, **Fig 3D**). Once intimal thickening formed, there was no difference in collagen compositions as detected by Sirius red stain in *Rac1$^{fl/fl}$/LC* and *Rac1$^{fl/fl}$* aortas (p>0.05, **S3 Fig in S1 File**).

### Secretion of inflammatory cytokines is reduced in RAC1-deficient macrophages

Reduced secretion of IL-6 (5.2 ± 0.2 ng/ml *versus* 6.8 ± 0.3 ng/ml, p<0.01, **Fig 4A**) as well as TNF-α (0.8 ± 0.08 ng/ml *versus* 1.0 ± 0.04 ng/ml, p<0.05, **Fig 4A**) was observed in cultured

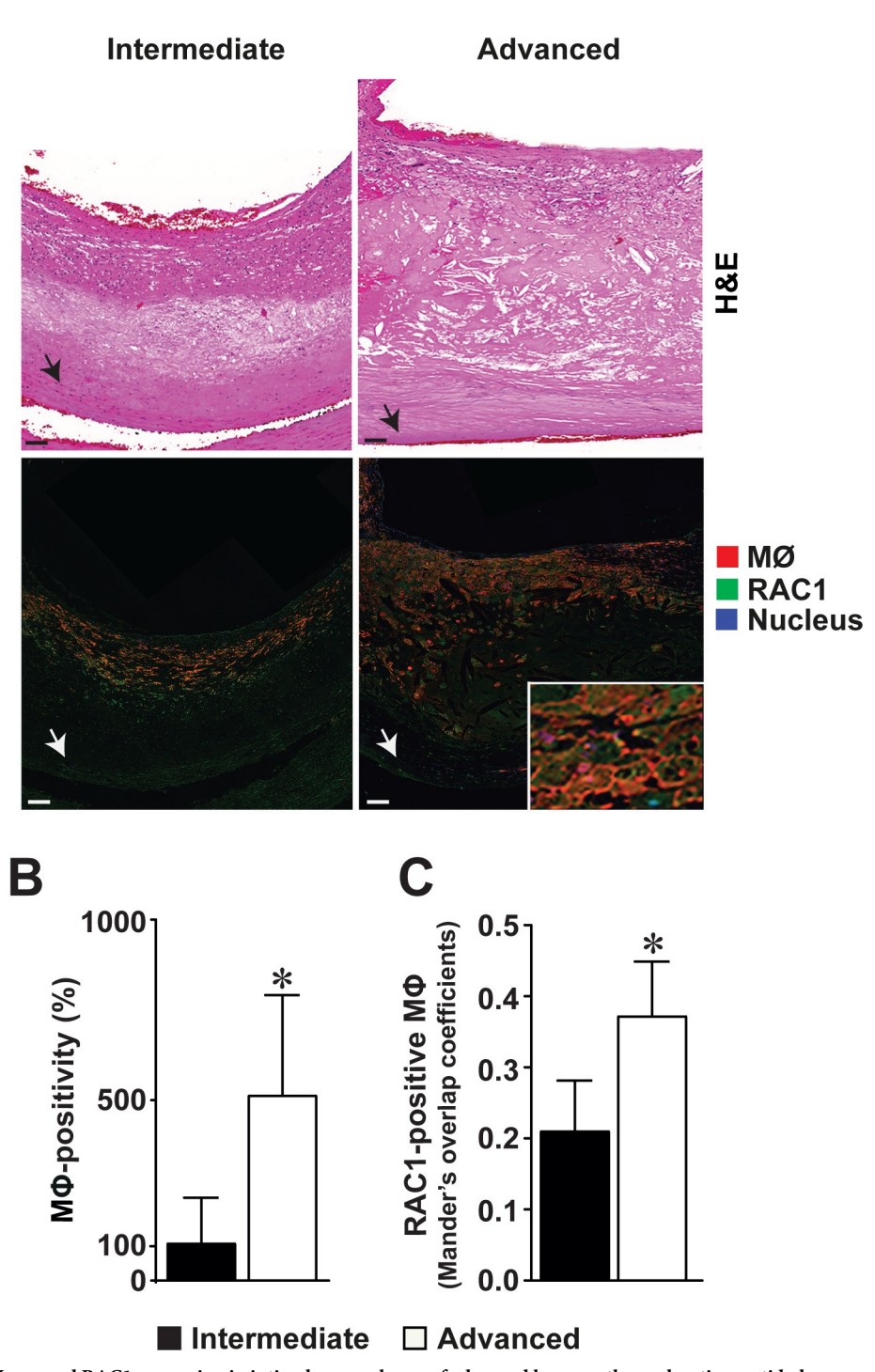

**Fig 1. Increased RAC1 expression in intimal macrophages of advanced human atherosclerotic carotid plaques.**
(**A**) Hematoxylin & Eosin-stained sections of intermediate and advanced atherosclerotic plaques in human carotid endarterectomies (upper panels). Immunofluorescence detection of RAC1 (green), and macrophages (red, MΦ) in the intermediate and advanced atherosclerotic plaques (lower panels). Nuclear staining by DAPI (blue). Arrowhead points to the internal elastic lamina bordering the intimal thickening from the medial layer. Scale bars represent 100 μm. Co-localization of RAC1 expression in MΦ in an advanced intimal thickening shown in enlarged inset. (**B**) Number of macrophages in intermediate and advanced atherosclerotic plaques. (**C**) Number of macrophages expressing RAC1 expression within the intimal thickening of intermediate and advanced atherosclerotic plaques. Mean ± SEM values (n = 9 in each). Wilcoxon signed-rank test was used. *p<0.05.

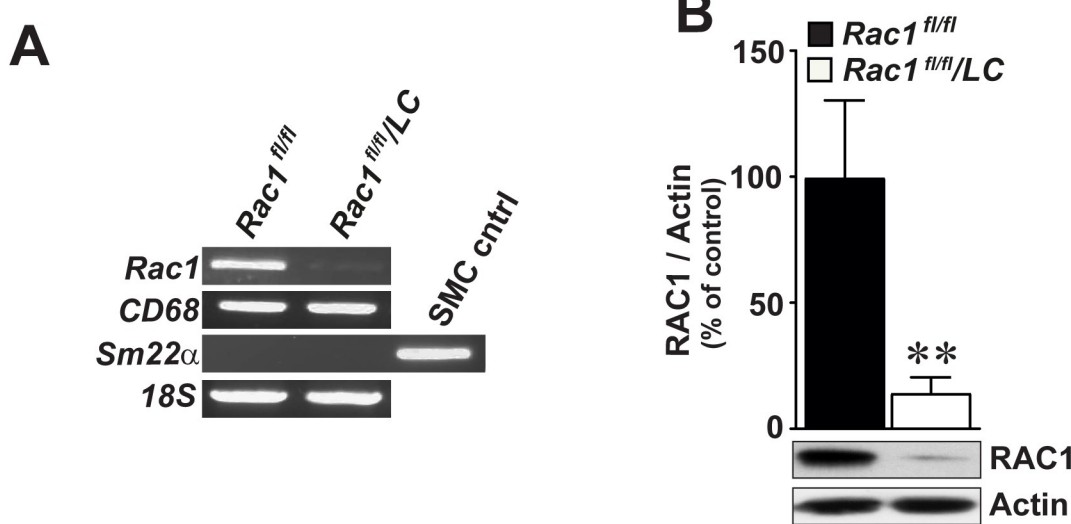

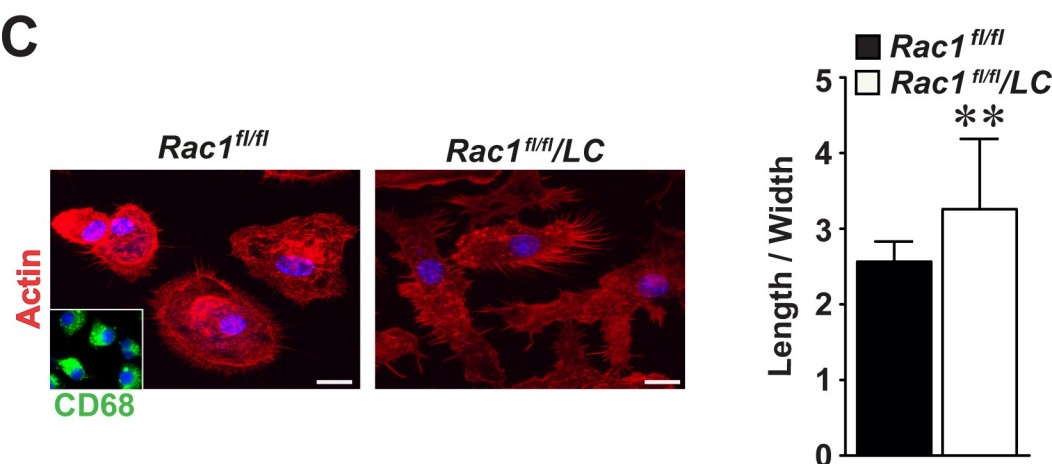

**Fig 2. Bone marrow-derived macrophages deficient for RAC1 display longer cell shape.** (**A**) RT-PCR analysis of *Rac1*, *CD68*, *and Sm22α* in cultured BMMs by gel electrophoresis. *18S* mRNA serves as an internal loading control. (**B**) Immunoblot analysis of RAC1 in *Rac1*$^{fl/fl}$ and *Rac1*$^{fl/fl}$/*LC* BMMs. Actin was included as an internal loading control. (**C**) Morphological shape of *Rac1*$^{fl/fl}$ and *Rac1*$^{fl/fl}$/*LC* BMMs as detected by actin immunofluorescence staining (left images). Inset demonstrates CD68-positivity in BMMs. Scale bars represent 10 μm. Quantification of ratios of macrophage cell elongation (right graphs). Mean ± SEM values of percentage or fold changes in at least triplicated data. Student's *t*-test was used. **p<0.01.

*Rac1*$^{fl/fl}$/*LC* BMMs compared to *Rac1*$^{fl/fl}$ controls. Similarly, serum levels of IL-6 were lower in atherogenic *Rac1*$^{fl/fl}$/*LC* mice compared to *Rac1*$^{fl/fl}$ control mice as detected by ELISA (0.40 ± 0.08 ng/ml *versus* 0.71 ± 0.05 ng/ml, p<0.05, **Fig 4B**). However, serum levels of TNF-α did not reach statistical significance in *Rac1*$^{fl/fl}$/*LC* mice (0.20 ± 0.09 ng/ml *versus* 0.31 ± 0.17 ng/ml, p>0.05, **Fig 4B**). IL-10 and IL-12 levels were not changed between *Rac1*$^{fl/fl}$/*LC* and *Rac1*$^{fl/fl}$ BMMs and serum levels of these cytokines were not detectable in *Rac1*$^{fl/fl}$/*LC* and *Rac1*$^{fl/fl}$ mice.

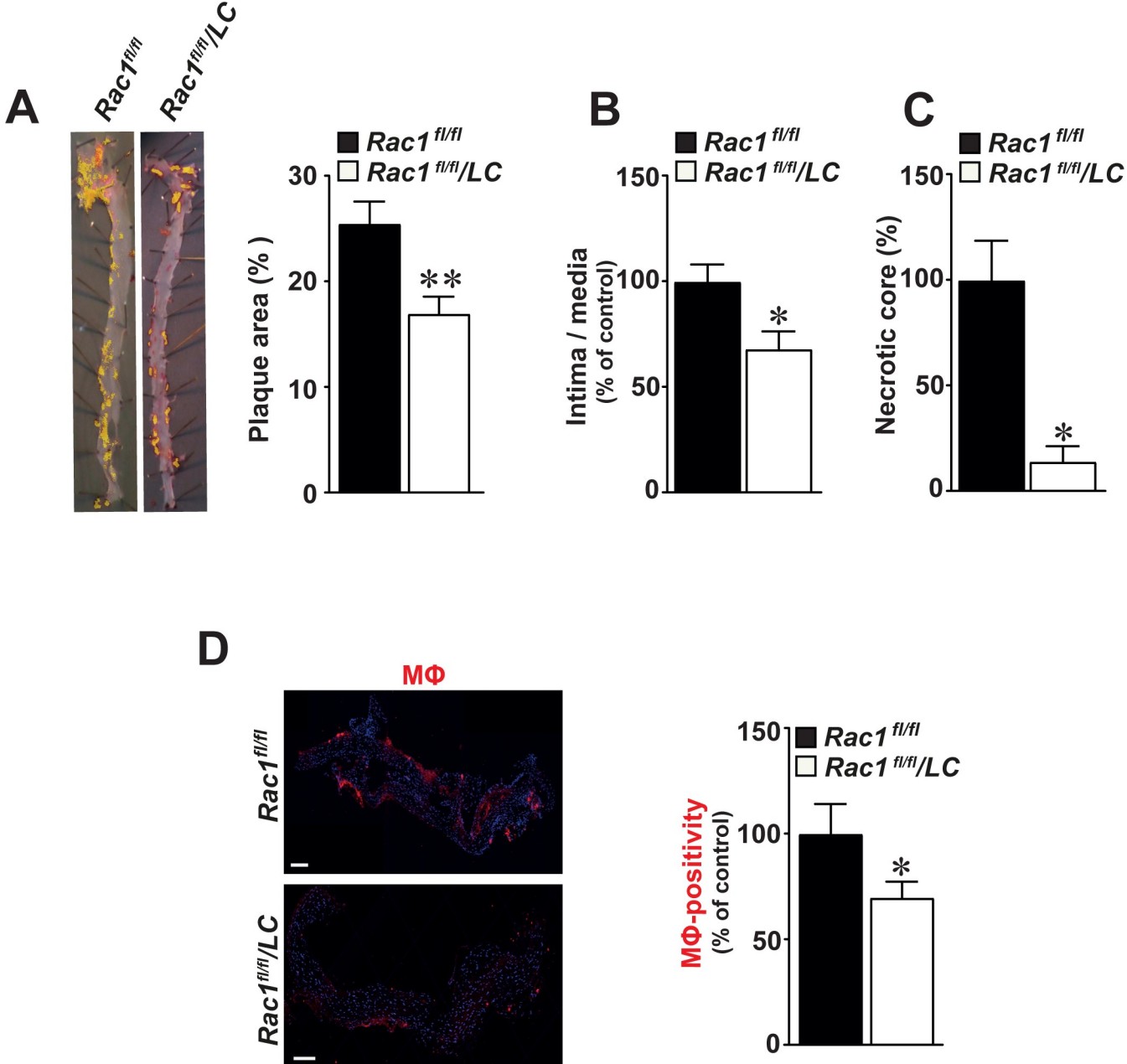

**Fig 3. Mice lacking RAC1 in macrophages develop smaller aortic atherosclerotic plaques.** (**A**) Representative images of Sudan IV-stained atherosclerotic aortic plaques in aortas obtained from *Rac1^fl/fl^*/LC mice infected with AdPCSK9 (n = 12) as compared to *Rac1^fl/fl^* control mice infected with AdPCSK9 (n = 12) (left panels). Quantification of atherosclerotic plaque size by image analysis in *Rac1^fl/fl^* and *Rac1^fl/fl^*/LC aortas infected with AdPCSK9 (right graphs). (**B**) Intima/media ratios in *Rac1^fl/fl^* and *Rac1^fl/fl^*/LC aortic arches from mice infected with AdPCSK9. (**C**) Percentage of intimal necrotic core areas in *Rac1^fl/fl^* and *Rac1^fl/fl^*/LC aortic arches. (**D**) Immunofluorescent detection of macrophages (MΦ) using anti-CD68 antibodies in *Rac1^fl/fl^* and *Rac1^fl/fl^*/LC aortic arches from mice infected with AdPCSK9 (left panels). Number of MΦ in *Rac1^fl/fl^* and *Rac1^fl/fl^*/LC aortic arches (right graphs). Scale bars represent 100 μm. Mean ± SEM values. Student's *t*-test was used. *p≤0.05.

## Deficiency of RAC1 in macrophages decreases lipid uptake

Compared to *Rac1^fl/fl^* control mice, *Rac1^fl/fl^*/LC mice produced higher serum levels of triglyceride, particularly in the VLDL/remnants and LDL-particles (**Fig 5A, S4 Fig in S1 File**). Compared to cultured *Rac1^fl/fl^* BMMs, *Rac1^fl/fl^*/LC BMMs displayed lower levels of mmLDL by 26%

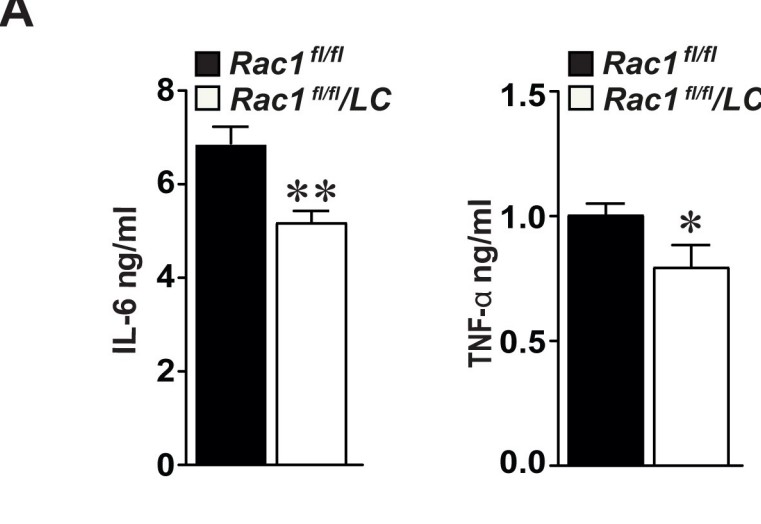

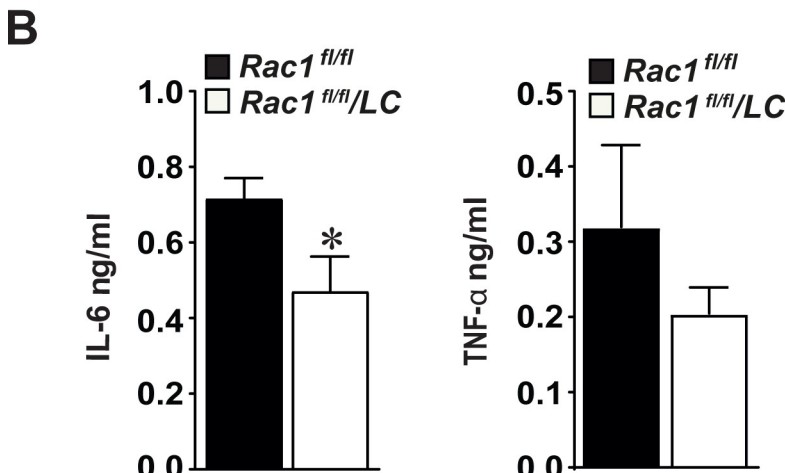

**Fig 4. Secretion of inflammatory cytokines is reduced in bone marrow-derived macrophages deficient for RAC1.**
(**A**) Secretion of IL-6 and TNF-α in primary $Rac1^{fl/fl}$ or $Rac1^{fl/fl}/LC$ BMMs as detected by ELISA (n = 6 in each) (**B**)
Serum blood levels of secreted IL-6 and TNF-α in atherogenic $Rac1^{fl/fl}$ or $Rac1^{fl/fl}/LC$ mice (n = 6 in each).
Mean ± SEM values. Student's $t$-test was used. *p<0.05 and **p<0.01.

(p<0.05, **Fig 5B**). As compared to $Rac1^{fl/fl}$ BMMs, $Rac1^{fl/fl}/LC$ BMMs displayed reduced
uptake of dil-labeled OxLDL after 8 hours (4.29 ± 0.29 *versus* 7.09 ± 0.92, p<0.05, **Fig 5C**).

We then assessed the expression levels of proteins that are involved in cholesterol metabolism using immunoblotting. Increased levels of CD36 (by 67%, p<0.05) and decreased levels
of both COX2 (by 45%, p<0.05), and ABCG1 (by 71%, $P$<0.05) were observed in $Rac1^{fl/fl}/LC$
BMMs compared to control $Rac1^{fl/fl}$ BMMs (**Fig 5D**). However, no difference was observed in
the expression of SR-B1, LXRα/β, and ABCA1 proteins.

## RAC1 interacts with FLNA in macrophages

As cell polarity is regulated by the organization of cytoskeleton, partly by actin-binding proteins, and activity of RAC1 is reduced in macrophages deficient for FLNA [12], we first

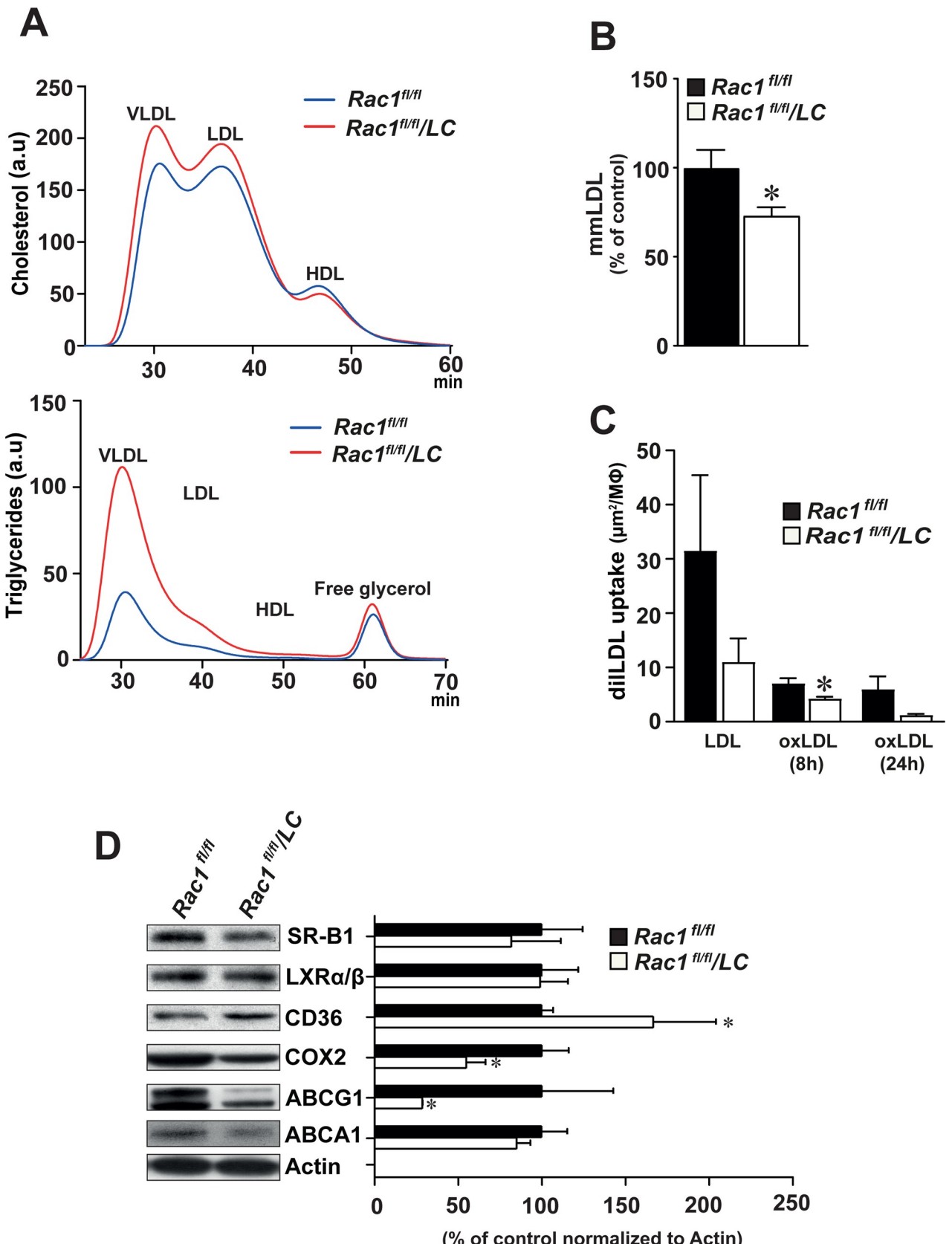

**Fig 5. Deficiency of RAC1 in bone marrow-derived macrophages decreases lipid uptake.** (**A**) Blood levels of cholesterol and triglyceride in fast protein liquid chromatography–fractionated plasma pooled from $Rac1^{fl/fl}$ mice (n = 5) that are infected with AdPCSK9 as compared to $Rac1^{fl/fl}/LC$ mice (n = 5) that are infected with AdPCSK9 after a high-fat diet for 24 weeks. HDL, high-density lipoprotein; IDL, intermediate-density lipoprotein; LDL, low-density lipoprotein; and VLDL, very-low-density lipoprotein. (**B**) Levels of minimally-modified LDL (mmLDL) in $Rac1^{fl/fl}$ and $Rac1^{fl/fl}/LC$ BMMs. Student's $t$-test was used. Mean ± SEM values. (**C**) Uptake of total LDL and oxidized LDL (OxLDL) for 8 and 24 hours in $Rac1^{fl/fl}$ and $Rac1^{fl/fl}/LC$ BMMs. Multiple groups were statistically evaluated by ANOVA and the Tukey-Kramer modification of Tukey's test and comparisons between two groups were evaluated by Student's $t$ test. (**D**) Representative Immunoblots and densitometry reading of SR-B1, LXRα/β, CD36, COX2, ABCG1, and ABCA1 bands in $Rac1^{fl/fl}$ and $Rac1^{fl/fl}/LC$ BMMs. Mean ± SEM values of triplicated data. Student's $t$-test was used. *p<0.05.

performed co-localization studies using immunofluorescently-labeled antibodies against RAC1 and FLNA. These data indicated that RAC1 and FLNA are co-expressed mainly in the cytoplasm of cultured BMMs (**Fig 6A**). Co-immunoprecipitation assay showed that RAC1 protein interacts physically with FLNA and *vice versa* (**Fig 6B**). In the absence of RAC1, we observed reduced levels of the cleaved C-terminal fragment of FLNA (FLNA^CT by 54%, p<0.01, **Fig 6C**). We also detected reduced levels of nuclear FLNA^CT expression in BMMs using immunofluorescence staining (**Fig 6D**, left panel) and quantified by image analysis (**Fig 6D**, right graph).

## Discussion

This study is the first to show that lack of RAC1 in macrophages *in vivo* inhibits aortic plaque size in an *in vivo* model of atherosclerosis. Macrophage functions including secretion of interleukins and foam cell formation are key events during the development of atherosclerosis. In the absence of RAC1, macrophages displayed impaired secretion of proinflammatory cytokines, reduced lipid uptake and foam cell formation. Furthermore, cytoskeletal FLNA^CT produced by calpain cleavage physically interacted with RAC1 and we have previously shown that lack of FLNA reduces macrophage activity and atherosclerosis [12]. These new insights not only provide an evident link between the biological role of RAC1 and atherogenesis, but also identify targets that may modify atherosclerotic plaque formation.

Studies in macrophages identified RAC1 as a regulator of the organization of actin cytoskeleton [24]. In our study, RAC1-deficiency in macrophages resulted in elongated cell shape. Besides, it is known that RAC1-deficiency reduces filopodial formation, but filopodial structures that are formed display increased extension [25]. Similarly, our images confirm this previous finding as RAC1-deficiency in macrophages leads to longer filopodial structures, but in less number. In addition to cytoskeletal functions, RAC1 regulates adhesion and migration [26], cell spreading and ruffling [25], phagocytosis [27], and transcriptional factors [28]. These pieces of evidence support a key role for RAC1 in many aspects of atherosclerotic plaque development. In this study, we observed an increased number of RAC1-expressing macrophages in advanced human carotid atherosclerotic plaques. As the knowledge on these pathophysiological processes modulated by macrophage-specific RAC1 is scanty and sometime contradictory, we initiated *in vitro* and *in vivo* experimental studies. To define whether elevated RAC1 expression in human atherosclerotic plaques may have a biological role in macrophages, we induced atherosclerosis in mice deficient for RAC1 in macrophages and observed fewer macrophages as well as smaller atherosclerotic plaque size. RAC1-deficiency in macrophages did not alter collagen composition, but reduced the size of necrotic cores within the atherosclerotic plaques.

Macrophages secrete multiple pro- and anti-inflammatory cytokines, as well as, depend on diverse stimuli. Currently, data points to interleukins and TNF-α as cytokines that, at least in some settings, are effective targets to reduce cardiovascular disease progression [29]. We observed decreased levels of the proinflammatory cytokine IL-6 and TNF-α in *Rac1*-deficient

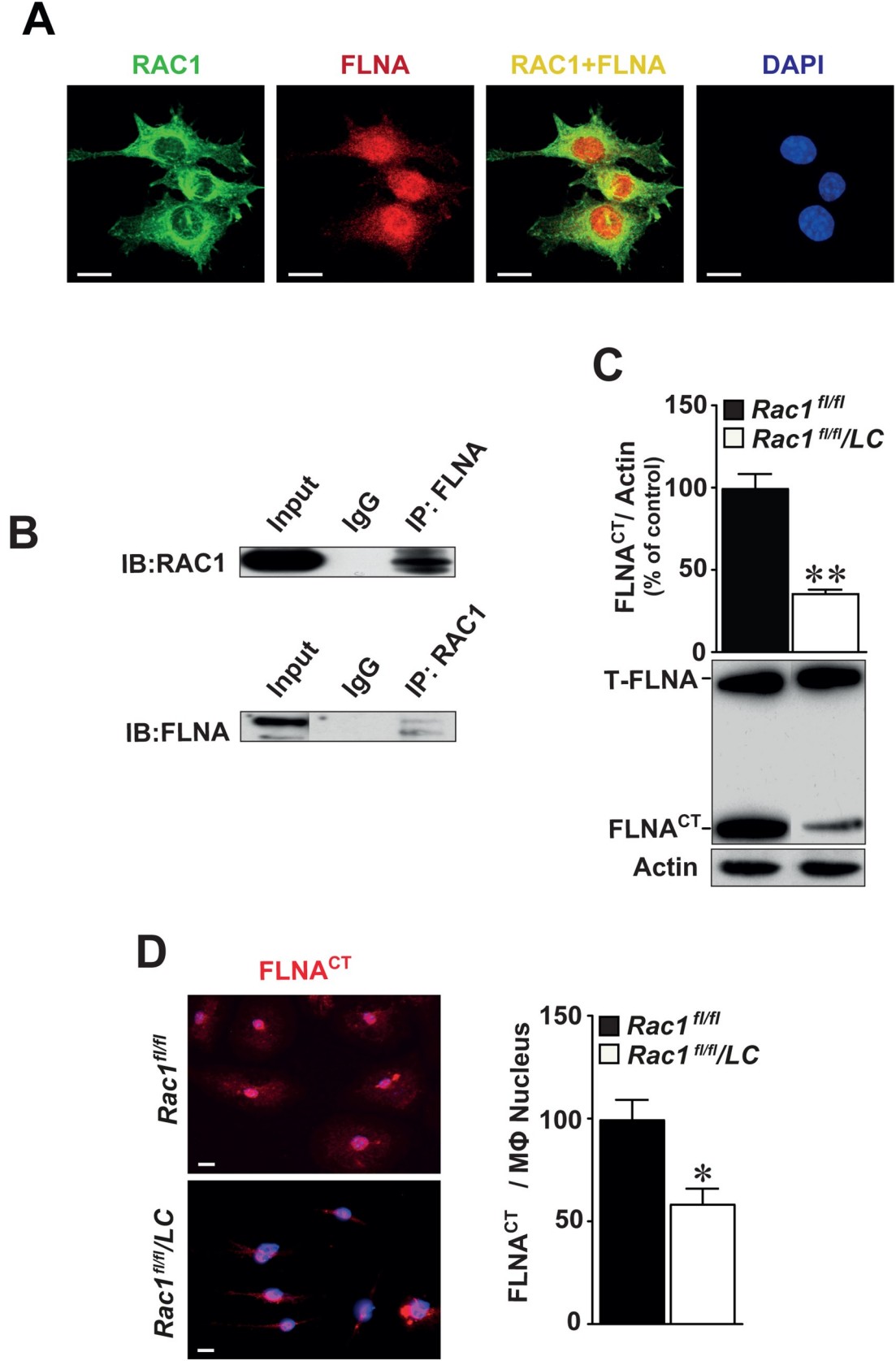

**Fig 6. RAC1 interacts with FLNA in the cytoplasm and deletion of RAC1 reduces the production of cleaved C-terminal fragment of filamin A (FLNA$^{CT}$).** (**A**) Expression of RAC1 (green) and FLNA (red) is co-localized mainly in the cytoplasm of BMMs (yellow) as detected by immunofluorescence staining. Scale bar represents 10 μm. (**B**) Co-immunoprecipitation identifying FLNA$^{CT}$ as an interacting partner of RAC1. Total proteins obtained from BMMs immunoprecipitated with FLNA$^{CT}$ and then immunoblotted with RAC1 antibodies (upper) or *vice versa* (lower). IgG served as negative controls. Full-length of FLNA and FLNA$^{CT}$ were indicated and actin served as internal loading control. (**C**) Immunoblotting of FLNA in *Rac1$^{fl/fl}$* and *Rac1$^{fl/fl}$/LC* BMMs. (**D**) Immunofluorescence staining of FLNA$^{CT}$ in *Rac1$^{fl/fl}$* and *Rac1$^{fl/fl}$/LC* BMMs. Quantification of nuclear FLNA$^{CT}$ expression in *Rac1$^{fl/fl}$* or *Rac1$^{fl/fl}$/LC* BMMs. Scale bar represents 20 μm. Mean ± SEM values of at least quadruplicated experiments. Student's *t*-test was used. *p<0.05, **p<0.01.

BMMs. It has been shown that IL-6 secretion is regulated via sphingosine-1-phosphate receptor 2 involving RHO/RHO-kinase and RAC1 signaling pathways [30]. Furthermore, studies using protein kinase inhibitors and dominant negative constructs demonstrate phosphatidylinositol 3-kinase/RAC1/PAK/JNK and phosphatidylinositol 3-kinase/RAC1/PAK/p38 pathways contribute to important roles in the late stages of TNF-α down-regulation of macrophage scavenger receptor expression [31]. We have earlier reported that both FLNA deficiency and inhibition of cleavage of FLNA$^{CT}$ by calpeptin results in reduced macrophage cell migration and proliferation [12]. However, RAC1-deficient BMMs expressing reduced levels of FLNA$^{CT}$ did not show differences in macrophage cell proliferation or migration in the present study.

RAC1 activation induced by free cholesterol accumulation in the plasma membrane is partly due to the activity of the membrane transporters, which modulates plasma membrane cholesterol content and lipid organization and, in the absence of an efficient extracellular acceptor, may cause cholesterol accumulation. The uptake and degradation of matrix-bound lipoproteins by macrophages require Rho family GTPases [24]. We found that *Rac1*-deficient BMMs display impaired lipid uptake as well as altered levels of CD36, COX2, and ABCG1. The fact that RAC1-deficient macrophages showed a decreased uptake of oxLDL despite elevated CD36 protein expression seems to be counterintuitive since CD36 has been reported to be an oxLDL receptor [32]. However, our experiments were performed in the absence of fatty acids which are important for binding of oxLDL to CD36 [23, 33]. The finding on the interaction between oxidized oxLDL and CD36 inducing loss of macrophage polarity and inhibiting macrophage locomotion [34], supports our observation on elevated CD36 expression in *Rac1*-deficient BMMs with impaired polarity and cell shape. Macrophage activation during motility increases COX2 protein levels [35]. Interestingly, our study showed that RAC1 deficiency in BMMs decreases COX2 levels. Furthermore, ABCG1 is robustly upregulated in macrophages taken from obese mice and regulates macrophage cholesterol levels [36]. Loss of ABCG1 from hematopoietic cells results in smaller atherosclerotic lesions populated with increased apoptotic macrophages [37]. Similarly, we found decreased levels of ABCG1 in *Rac1*-deficient BMMs that show decreased lipid uptake, which resulted in smaller fatty atherosclerotic plaques.

Once retention of LDL takes place in the subendothelium, the internalization and degradation of matrix-retained and aggregated LDL by macrophages may involve the actin-myosin cytoskeleton [24]. In addition, phagocytosis is the mechanism of internalization used by specialized cells such as macrophages and also requires macrophage shape changes and phenotypic polarization by actin filaments. Recently, we reported that RAC1 activity is reduced in FLNA-deficient BMMs [12]. In this study, we identified actin-binding protein FLNA as an interacting partner of RAC1 in BMMs. Interestingly, RAC1 activators and downstream effectors control the local reorganization of the actin cytoskeleton beneath bound particles during ingestion [24]. FLNA is required for differentiation of monocytes by regulating actin dynamics via Rho GTPases that control monocyte migration [24] and also may be a scaffold for the spatial organization of Rho-GTPase-mediated signaling pathways [11]. In the absence of FLNA, levels of active RAC1 are reduced [18]. Human FLNA mutations associated with valvular

dystrophy alter the balance between RHOA and RAC1 GTPases activities in favor of RHOA, providing evidence for a role of the RAC1-specific GTPase activating protein FilGAP, a RAC-specific Rho-GTPase-activating protein [38]. Furthermore, we observed lower levels of cleaved FLNA$^{CT}$ and its nuclear expression in *Rac1*-deficient BMMs in this study. We have recently reported that genetic inactivation of FLNA and chemical inhibition of calpain-dependent cleavage of FLNA impairs macrophage signaling and function, and reduces atherosclerosis in mice [12]. Thus, this way of chemical inhibition would also be administered to target RAC1 signaling.

Overall, our study reporting RAC1-dependent cytokine release and lipid biology in macrophages and FLNA as an interacting partner of RAC1 may indicate new potential mechanisms behind atherogenesis *in vivo*. The finding of increased RAC1 expression in human macrophages within advanced carotid artery plaques suggests that RAC1 expression can be a prognostic biomarker for atherogenesis. Besides, RAC1 represents an attractive therapeutic target for cardiovascular diseases; however, the clinical search for effective RAC1 inhibitors is still underway [4]. Nevertheless, characterizing these novel mechanisms provide invaluable information regarding major macrophage events that are mediated by aberrant RAC1 signaling. Importantly, our results can be utilized to further facilitate the development of effective pharmacological agents that can inhibit RAC1 signaling in macrophages in atherosclerosis-related cardiovascular disorders.

## Supporting information

**S1 File.**
(PDF)

## Acknowledgments

The authors thank Kristina Skålen (The Wallenberg Laboratory, University of Gothenburg, Sweden), Shahin de Lara and Samad Parhizkar (Department of Clinical Pathology, Sahlgrenska University Hospital, Sweden) for excellent technical help and Toshima Parris (University of Gothenburg, Sweden) for editing the manuscript.

## Author Contributions

**Conceptualization:** Sashidar Bandaru, Levent M. Akyürek.

**Data curation:** Sashidar Bandaru.

**Formal analysis:** Sashidar Bandaru.

**Funding acquisition:** Levent M. Akyürek.

**Investigation:** Sashidar Bandaru, Chandu Ala, Matias Ekstrand, Murali K. Akula, Matteo Pedrelli, Xi Liu, Liliana Håversen, Levent M. Akyürek.

**Methodology:** Sashidar Bandaru.

**Project administration:** Sashidar Bandaru, Levent M. Akyürek.

**Resources:** Göran Bergström, Jan Borén, Martin O. Bergo, Levent M. Akyürek.

**Software:** Sashidar Bandaru.

**Supervision:** Levent M. Akyürek.

**Validation:** Sashidar Bandaru.

**Visualization:** Sashidar Bandaru.

**Writing – original draft:** Sashidar Bandaru, Levent M. Akyürek.

**Writing – review & editing:** Levent M. Akyürek.

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
