## [Decision Letter · Decision Letter 0]

19 Mar 2020

PONE-D-20-06077

Lack of RAC1 in macrophages protects against atherosclerosis

PLOS ONE

Dear Dr Ayürek,,

Thank you for submitting your manuscript to PLOS ONE. After careful consideration, we feel that it has merit but does not fully meet PLOS ONE’s publication criteria as it currently stands. Therefore, we invite you to submit a revised version of the manuscript that addresses the points raised during the review process.

Please answer all the questions or reviewers 1 and 2.

We would appreciate receiving your revised manuscript by July 2020. To enhance the reproducibility of your results, we recommend that if applicable you deposit your laboratory protocols in protocols.io, where a protocol can be assigned its own identifier (DOI) such that it can be cited independently in the future. For instructions see: http://journals.plos.org/plosone/s/submission-guidelines#loc-laboratory-protocols

We look forward to receiving your revised manuscript.

Kind regards,

Esther Lutgens

Academic Editor

PLOS ONE

Journal Requirements:

Reviewers' comments:

Reviewer's Responses to Questions

**Comments to the Author**

1. Is the manuscript technically sound, and do the data support the conclusions?

Reviewer #1: Partly

Reviewer #2: Yes

2. Has the statistical analysis been performed appropriately and rigorously? 

Reviewer #1: Yes

Reviewer #2: Yes

3. Have the authors made all data underlying the findings in their manuscript fully available?

Reviewer #1: No

Reviewer #2: Yes

4. Is the manuscript presented in an intelligible fashion and written in standard English?

Reviewer #1: No

Reviewer #2: Yes

5. Review Comments to the Author

Reviewer #1: The manuscript of Akyürek and colleagues describes the interesting finding of decreased atherogenesis upon macrophage-specific deletion of RAC1. The authors show an increased expression of RAC1 in macrophages in human advanced plaques and demonstrate that deletion of RAC1 in (mouse) macrophages affects their shape and function. Differences in shape are explained by interactions between RAC1 and FLNA to remodel the cytoskeleton for migration and lipid uptake. Mice with RAC1-deficient macrophages show decreased plaque size and decreased macrophage content in the plaques. In vitro, isolated macrophages show decreased inflammatory activation by decreased cytokine secretion and decreased lipid uptake. The data described in this manuscript are potentially of significance for clinical application and in the identification of novel therapeutic targets.

However, some key information seems to be missing:

1. First, in the in vitro experiments cultured macrophages are used to validate in vivo findings. However, the culturing and activation of these cells is not clearly described. Concerning the isolation, the material and method section refers to a paper in which only the culturing is explained. Throughout the result section it is not always clear if the authors are talking about cultured BMMs or about lesional macrophages nor what they mean with “extracted macrophages”. It should be clarified in the M&M, text, figures and legends which macrophages are being investigated and how they were stimulated (included, timing and concentrations). This crucial information is now missing. Moreover, untreated macrophages should be shown along to their LPS-treated ones.

2. The authors indicate purity of the cultured macrophages by showing a lack of SM22a mRNA in the culture. However, it would be more informative if the purity is validated by FACS staining for both CD11b and F4/80 as the absence of SM22a does not prove purity.

3. Authors state in the discussion that macrophages display impaired secretion of proinflammatory cytokines, but only show that some are impaired and some are not. Also, it is unclear how these macrophages are activated. What is the concentration and the timing of the LPS used in this study? It would be of added value to also plot the cytokine secretion of unstimulated macrophages. Where in vitro 2 out of 4 cytokines show a decrease in secreted cytokine levels, in the serum only 1 cytokine was measured. Are serum cytokine levels behaving similarly to cytokine secretion by cultured macrophages?

4. The authors show a decreased ability for lipid uptake in the RAC1 deficient macrophages, but do not show the implications of the found proteins in this pathway. In fact, the deregulation of the proteins appears in conflict to the observed changes in lipid uptake. This should at least be discussed.

5. In the final paragraph of the results, the mechanistic link between RAC1 and FLNA is shown, but the rationale of doing these experiments is missing in there. It is unclear what the importance and relevance of these results is and this becomes only somewhat apparent when reading the discussion. The results section should be improve in a way that readers understand why particular experiments were done and what the importance of the findings is.

6. In the last paragraph of the discussion, the authors state to have found a causative correlation between RAC1-dependent cytokine release in macrophages. This should be rephrased or should be functionally proven as the current layout of experiments does not allow to make such statements.

7. Figure 1 shows the positivity of macrophages in pixels. It would be of added value to plot the data per µm2 or as a ratio to the total amount of pixels.

8. Figure 3C shows a reduced macrophage-positivity in RAC1fl/fl/LC. How is this quantified? Is this per µm2 of plaque, or in the picture in general? Could smaller plaques explain reduced numbers of macrophages seen in this picture? This should be clarified.

9. Figure 6B (bottom) and C shows an immunoblot which has clearly been edited (cut and paste). are these data from different blots? This can only be allowed if the complete blots are provided as supplemental data, highlight the parts that were cut and used in the actual figures. Preferentially, a blot should be performed in which the conditions of interest are blotted next to each other. Otherwise, the protein levels cannot be directly compared.

Moreover, the text should be reviewed and edited by a native English speaker to correct difficult statements and sentences. For example: accompanied with -> accompanied by; “macrophages were more elongated compared to Rac1fl/fl control macrophages using immunofluorescence staining of actin phalloidin”; “Compared to Rac1fl/fl controls, higher serum levels of triglyceride, particularly in the VLDL/remnants and LDL-particles, were observed in Rac1fl/fl/LC mice (Fig 5A).”; “macrophages displayed lower levels of mmLDL by 26% compared with Rac1fl/fl macrophages”, etc.

Reviewer #2: Bandaru et al Pone 2020

The manuscript by Bandaru et al. describes the presence of RAC1 in macrophages and smooth muscle cells in human atherosclerotic plaques, prompting them to investigate its role in experimental atherosclerosis and macrophages using a Lysosyme-driven Cre recombinase approach.

They show that mice lacking RAC1 in macorphages develop smaller lesions, and that the macrophages are less inflammatory, and decreased lipid uptake.

The manuscript seems technically sound.

A few issues need to be addressed.

1. Fig 2C. The authors describe that the macrophages have an altered length-to-width ratio. However, the KO macrophages also seem to develop more filopodia-like structures, which remains undiscussed

2. Fig. S2B. The authors measure macrophage migration. Please provide the rationale for measuring this parameter. In addition, the methods do not seem to be described.

3. Plaques of KO mice are smaller, however what are other plaque parameters such as severity, necrosis, collagen content,...

4. Fig3B. How was intima/media ratio in mice measured?

5. Fig. 3C: was this total macrophage area or relative to plaque size?

6. The authors describe the inflammatory parameters of the KO macrophages and in KO mice. Please clearly/explicitely describe what was measeured in vitro and what was measured in vivo. Was in vitro secretion after inflammatory stimulation or in basal conditions? What were the levels of TNF in vivo? And given the recent results in the Cantos trial, IL1b measurements could strengthen the paper

7. Fig. 5A. What were total serum cholesterol and TG levels?

8. Fig5B. The authors describe KO macrophages to display lower levels of mmLDL. How was this measured? Or do the authors mean reduced uptake of mmLDL?

9. The authors nicely show FLNA interaction in macrophages. Can they also provide evidence of reduced levels in the KO mice in vivo? And localization of FLNA in the human plaques?

6. PLOS authors have the option to publish the peer review history of their article (what does this mean?). If published, this will include your full peer review and any attached files.

Reviewer #1: No

Reviewer #2: Yes: Kristiaan Wouters

---

## [Author Response · Author response to Decision Letter 0]

31 May 2020

Reviewer #1:

1. First, in the in vitro experiments cultured macrophages are used to validate in vivo findings.However, the culturing and activation of these cells is not clearly described. Concerning the isolation, the material and method section refers to a paper in which only the culturing is explained. Throughout the result section it is not always clear if the authors are talking about cultured BMMs or about lesional macrophages nor what they mean with “extracted macrophages”. It should be clarified in the M&M, text, figures and legends which macrophages are being investigated and how they were stimulated (included, timing and concentrations). This crucial information is now missing. Moreover, untreated macrophages should be shown along to their LPS-treated ones.

Thanks for these comments and we do apologize for the confusion. In this study, we have not cultured lesional macrophages and we have presented assays using bone marrow macrophages (BMMs) exposed to LPS in all experimental groups. Thus, the wording “macrophages” within the context of our experiments has been replaced with “BMMs” throughout the manuscript including M&M, text body, figures, and figure legends as necessary. We have extracted bone marrow monocytes from both femur and tibia of the mice as described elsewhere (ref# 12). To differentiate monocytes into macrophages, bone marrow monocytes were cultured with DMEM medium containing 10% FBS, 1% glutamine, 1% gentamicin,1% HEPES, 0.01% β-mercaptoethanol, and 10% CMG14-12 cell line supernatant as a source of mouse M-CSF. One week later, BMMs were plated and assayed up to 3 days. We have now provided this information (page 7, lines 4–8). We stimulated BMMs with 10 ng/ml LPS 15 minutes before immunoblotting and 8 hours before ELISA (page 7, lines 20-21). As requested, additional experimental groups without LPS have been included in experimental setting. We tested untreated BMMs for ELISA (Reviewer Figure 1) as well as for immunoblotting of the cultured BMMs without LPS (Reviewer Figure 2), however, these assays did not result in significant differences. Please see the attached figure including these untreated BMMs along with LPS-treated BMMs to convince the Reviewer.

2. The authors indicate purity of the cultured macrophages by showing a lack of SM22a mRNA in the culture. However, it would be more informative if the purity is validated by FACS staining for both CD11b and F4/80 as the absence of SM22a does not prove purity. 

In addition to experimental proof of purity by mRNA expression, we have immunofluorescently stained cultured BMMs using a macrophage-specific CD68 antibody as reported earlier (ref# 12). An inset of image demonstrating CD68-positive BMMs has been included in Figure 2C. This information has been added to M&M (page 7, line 15-16) and figure legends.

3. Authors state in the discussion that macrophages display impaired secretion of proinflammatory cytokines, but only show that some are impaired and some are not. Also, it is unclear how these macrophages are activated. What is the concentration and the timing of the LPS used in this study? It would be of added value to also plot the cytokine secretion of unstimulated macrophages. Where in vitro 2 out of 4 cytokines show a decrease in secreted cytokine levels, in the serum only 1 cytokine was measured. Are serum cytokine levels behaving similarly to cytokine secretion by cultured macrophages?

In addition to cultured BMMs, we studied mouse blood serum levels of IL-10, IL-12, and TNF-α. However, no significant differences were seen in secretion of these cytokines between the experimental mouse groups. The only difference that we observed was the level of IL-6, which has been provided in Figure 4B. Blood levels of other cytokines in Rac1-deficient mice were not decreased as compared to in vitro experiments probably due to the dilutional effects omitting local changes in vivo. It seems that BMMs secrete high levels of IL-6 which seem to be enough to detect even in vivo. However, IL-10, IL-12, and TNF-α are secreted already at low levels as detected in vitro, making it difficult to measure them in vivo. Prior to the ELISA assays, BMMs were stimulated with 10 ng/ml LPS for 8 hours. For ELISA experiments, we also included untreated BMMs for comparison with LPS-stimulated BMMs, but comparisons of these experimental groups did not result in significant differences. As mentioned above, we have now provided these data to convince you without including it in the manuscript (Reviewer Figures 1 and 2).

4. The authors show a decreased ability for lipid uptake in the RAC1 deficient macrophages, but do not show the implications of the found proteins in this pathway. In fact, the deregulation of the proteins appears in conflict to the observed changes in lipid uptake. This should at least be discussed. 

This issue had been initially discussed in the fourth paragraph of the Discussion. To clear this further, we have now provided the following argument in the Discussion. “The fact that RAC1-deficient macrophages showed a decreased uptake of oxLDL despite elevated CD36 protein expression seems to be counterintuitive since CD36 has been reported to be an oxLDL receptor [32]. However, our experiments were performed in the absence of fatty acids which are important for binding of oxLDL to CD36 [23, 33]” in (page 14, lines 12–16).

5. In the final paragraph of the results, the mechanistic link between RAC1 and FLNA is shown, but the rationale of doing these experiments is missing in there. It is unclear what the importance and relevance of these results is and this becomes only somewhat apparent when reading the discussion. The results section should be improve in a way that readers understand why particular experiments were done and what the importance of the findings is.

In this study, macrophages deficient for RAC1 displayed cell shape changes (Figure 2C and 2D). Cell polarity is regulated by organization of cytoskeleton, partly by actin-binding proteins including FLNA. We have recently reported that activity of RAC1 is reduced in macrophages deficient for FLNA (ref# 12). To provide some insights into mechanisms behind the reduced atherosclerotic plaque formation as a result of RAC1 deficiency in macrophages, we determined if RAC1 physically interacts with FLNA. In addition, RAC1-deficient macrophages express lower levels of the C-terminal fragment of FLNA (FLNACT) that is produced by calpains, providing new tools to modulate these mechanisms by chemicals. These motivations have been included in the Results (page 12, lines 2-3) and Discussion (page 14, lines 3–6) as well as importance of this interaction in the Discussion (page 15, lines 14–16).

6. In the last paragraph of the discussion, the authors state to have found a causative correlation between RAC1-dependent cytokine release in macrophages. This should be rephrased or should be functionally proven as the current layout of experiments does not allow to make such statements.

We do agree with this criticism and have rephrased this sentence as suggested in the Discussion.

7. Figure 1 shows the positivity of macrophages in pixels. It would be of added value to plot the data per μm2 or as a ratio to the total amount of pixels.

In our images, one μm2 contains 13.2 pixels. Instead of converting the pixel data to μm2 in this panel (Figure 1B), we have now presented it as percentages to keep its comparative presentation consistent with all other figures.

8. Figure 3C shows a reduced macrophage-positivity in RAC1fl/fl/LC. How is this quantified? Is this per μm2 of plaque, or in the picture in general? Could smaller plaques explain reduced numbers of macrophages seen in this picture? This should be clarified.

In Figure 3C, macrophage-positivity detected by immunofluorescence staining was quantified in the entire area of the aortic vessel wall. The total number of pixels have been normalized to the control group and differences were presented by percentages. This information has been added to the M&M (page 8, lines 9-10).

9. Figure 6B (bottom) and C shows an immunoblot which has clearly been edited (cut and paste). are these data from different blots? This can only be allowed if the complete blots are provided as supplemental data, highlight the parts that were cut and used in the actual figures. Preferentially, a blot should be performed in which the conditions of interest are blotted next to each other. Otherwise, the protein levels cannot be directly compared.

As required, we have provided the original unadjusted and uncropped images for all gel data and immunoblots included in this manuscript as supporting information (S4 Fig). As requested, the protein bands that were cut and used in the actual figures have been highlighted in rectangular red frames for convenience.

Moreover, the text should be reviewed and edited by a native English speaker to correct difficult statements and sentences. For example: accompanied with -> accompanied by; “macrophages were more elongated compared to Rac1fl/fl control macrophages using immunofluorescence staining of actin phalloidin”; “Compared to Rac1fl/fl controls, higher serum levels of triglyceride, particularly in the VLDL/remnants and LDL-particles, were observed in Rac1fl/fl/LC mice (Fig 5A).”; “macrophages displayed lower levels of mmLDL by 26% compared with Rac1fl/fl macrophages”, etc.

These sentences have been rephrased as requested and the entire manuscript text has been reviewed by a native English-speaking scientist.

Reviewer #2:

1. Fig 2C. The authors describe that the macrophages have an altered length-to-width ratio. However, the KO macrophages also seem to develop more filopodia-like structures, which remains undiscussed.

In this study, ratios of macrophage length-to-width have been given without analyses of filopodia-like structures. As pointed out, it is known that RAC1-deficiency increases extension of filopodial formation; however, filopodial formation was reduced. This finding has been discussed and the reference (ref# 25) has been indicated in the Discussion (page 13, lines 12–14).

2. Fig. S2B. The authors measure macrophage migration. Please provide the rationale for measuring this parameter. In addition, the methods do not seem to be described.

We have earlier reported that both FLNA deficiency and inhibition of cleavage of FLNACT by calpeptin results in reduced macrophage cell migration (ref # 12). In this study, we observed that RAC1 deficiency reduces FLNACT levels (Figure 6C). Thus, we questioned if reduced levels of FLNACT in RAC1-deficient macrophages could also reduce macrophage cell migration. However, RAC1 deficiency in macrophages did not alter migration (S2B Fig). This information has been added to the Discussion (page 14, lines 2). BMMs were assayed for migration using modified Boyden chambers up to 18 hours as described earlier (ref# 12). This information has been added to the M&M (page 7, last two lines).

3. Plaques of KO mice are smaller, however what are other plaque parameters such as severity, necrosis, collagen content,...

We have analyzed cellular composition including macrophages and vascular smooth muscle cells within the atherosclerotic plaques and quantified lipid content as we considered it as the most relevant parameter; nevertheless, severity of plaque type, necrosis or fibrosis are also interesting features of plaques as pointed out. Hopefully, we will include these parameters in future more detailed histopathological studies.

4. Fig3B. How was intima/media ratio in mice measured?

As reported earlier (ref # 12, page 8, lines 12-13), paraffin-embedded mouse aortic arches were sectioned near the aortic valves and stained with H&E. Images of sectioned aortas were scanned using the Mirax Scanner (Zeiss, Germany). To measure intima/media ratios, surface areas of intimal and medial layers were outlined manually in captured images and quantified separately by BioPix iQ.

5. Fig. 3C: was this total macrophage area or relative to plaque size?

In Figure 3C, macrophage-positivity detected by immunofluorescence staining was quantified in the entire area of aortic vessel wall. This information has been added to the M&M (page 8, lines 9-10).

6. The authors describe the inflammatory parameters of the KO macrophages and in KO mice. Please clearly/explicitely describe what was measeured in vitro and what was measured in vivo. Was in vitro secretion after inflammatory stimulation or in basal conditions? What were the levels of TNF in vivo? And given the recent results in the Cantos trial, IL1b measurements could strengthen the paper. 

In BMMs, secreted levels of IL-6, IL-10, IL-12 and TNF-α were detected (Figures 4A and 4CE); however, only secreted levels of IL-6 were presented from mouse serum (Figure 4B). Cultured BMMs were stimulated by LPS. In this manuscript, we have not included data on basal BMMs without LPS stimulation, but we have now provided data on BMMs without LPS stimulation to convince the Reviewers in this letter (Reviewer Figure 1). In this study, TNF-α levels were presented only from cultured BMMs extracted from experimental mice. We have also analyzed this cytokine in vivo; however, no difference was measured. TNF-α secretion by intimal macrophages may also be increased; nevertheless, it seems to not be enough to detect in vivo, probably due to dilutional effects. We analyzed levels of IL-1β secretion both in cultured BMMs and experimental mice; however, we did not detect this cytokine either in vitro or in vivo.

7. Fig. 5A. What were total serum cholesterol and TG levels?

As requested, we have now included levels of total serum cholesterol and TG (S3 Fig). In the absence of RAC1, levels of total cholesterol were not altered, whereas increased levels of TG were detected in Rac1-deficient blood serum.

8. Fig5B. The authors describe KO macrophages to display lower levels of mmLDL. How was this measured? Or do the authors mean reduced uptake of mmLDL?

As reported earlier (ref# 12), BMMs were incubated with 50 μg/mL mmLDL. After 24 hours, intracellular levels of mmLDL were measured. BMMs were then fixed with ethanol and stained with Oil-Red-O. Images of BMMs were captured using a Zeiss microscope and intracellular staining were measured using BioPix iQ software. This assay was used for foam cell formation, not for mmLDL uptake.

9. The authors nicely show FLNA interaction in macrophages. Can they also provide evidence of reduced levels in the KO mice in vivo? And localization of FLNA in the human plaques?

In this study, we have not studied the level of FLNA in aortic atherosclerotic plaques. However, we would like to emphasize that reduced levels of FLNA expressed by cultured macrophages have been obtained from Rac1-deficient mice (Figure 6C), not from BMMs silenced for mRNA expression of Rac1. Nevertheless, we have earlier localized expression of FLNA to intimal macrophages in human carotid artery plaques (ref# 12, Figures 1C–1H). In addition to human macrophages, human intimal vascular smooth cells were also positive for FLNA expression.

---

## [Decision Letter · Decision Letter 1]

14 Jul 2020

PONE-D-20-06077R1

Lack of RAC1 in macrophages protects against atherosclerosis

PLOS ONE

Dear Dr. Akyürek,

Thank you for submitting your manuscript to PLOS ONE. After careful consideration, we feel that it has merit but does not fully meet PLOS ONE’s publication criteria as it currently stands. Therefore, we invite you to submit a revised version of the manuscript that addresses the points still raised by both reviewers.

We look forward to receiving your revised manuscript.

Kind regards,

Michael Bader

Academic Editor

PLOS ONE

Reviewers' comments:

Reviewer's Responses to Questions

**Comments to the Author**

1. If the authors have adequately addressed your comments raised in a previous round of review and you feel that this manuscript is now acceptable for publication, you may indicate that here to bypass the “Comments to the Author” section, enter your conflict of interest statement in the “Confidential to Editor” section, and submit your "Accept" recommendation.

Reviewer #1: (No Response)

Reviewer #2: (No Response)

2. Is the manuscript technically sound, and do the data support the conclusions?

Reviewer #1: Yes

Reviewer #2: Partly

3. Has the statistical analysis been performed appropriately and rigorously? 

Reviewer #1: Yes

Reviewer #2: Yes

4. Have the authors made all data underlying the findings in their manuscript fully available?

Reviewer #1: Yes

Reviewer #2: Yes

5. Is the manuscript presented in an intelligible fashion and written in standard English?

Reviewer #1: No

Reviewer #2: Yes

6. Review Comments to the Author

Reviewer #1: The authors addressed most of my and the other's reviewers’ concerns and in principal the manuscript is ready for publication after some minor, but important corrections.

1/ While the response letter suggests the authors included the naïve controls, this information is not included in the figures. This should be included in the figures of the paper. Also, I did not receive the “attached figure” mentioned.

“As requested, additional experimental groups without LPS have

been included in experimental setting. We tested untreated BMMs for ELISA (Reviewer Figure

1) as well as for immunoblotting (Reviewer Figure 2), however, these assays did not result in

significant differences. Please see the attached figure including these untreated BMMs along

with LPS-treated BMMs to convince the Reviewer.”

2/ The text should be copyedited by a professional to improve readability and correct mistakes. A few have been corrected but orthers remain. E.g. stimulated by LPS -> stimulated with LPS.

Reviewer #2: The authors have attempted to improve the manuscript by clarifying some issues in the text. Moreover, they performed an additional measurement of IL1b in vitro and in vivo in addition to describing TNF protein levels in vivo.

Although these efforts have certainly improved the quality of the manuscript, some issues remain.

1. The authors state that “ it is known that RAC1-deficiency increases extension of filopodial formation; however, filopodial formation was reduced”. However, on the picture show it seems that filopodia are increased in the KO. Does this mean that the picture presented in Fig. 2C is not representative?. Moreover, the text added in the discussion does not clarify the observation to me.

2. The authors state that they hopefully will perform more detailed analysis of plaque parameters in the future. However, given that they have paraffin embedded slides and HE stainings, I see no reason why it would not be possible to do a scoring of necrotic core on these sections and one additional Sirius red staining (which is relatively simple to perform).

3. It remains unclear whether IL10 and IL12 were not measured in vivo or not presented. I would also argue to add the measurements of TNF in vivo to the manuscript and to discuss these.

7. PLOS authors have the option to publish the peer review history of their article (what does this mean?). If published, this will include your full peer review and any attached files.

Reviewer #1: No

Reviewer #2: No

---

## [Author Response · Author response to Decision Letter 1]

11 Aug 2020

Reviewer #1: 

1/ While the response letter suggests the authors included the naïve controls, this information is not included in the figures. This should be included in the figures of the paper. Also, I did not receive the “attached figure” mentioned.

“As requested, additional experimental groups without LPS have been included in experimental setting. We tested untreated BMMs for ELISA (Reviewer Figure 1) as well as for immunoblotting (Reviewer Figure 2), however, these assays did not result in significant differences. Please see the attached figure including these untreated BMMs along with LPS-treated BMMs to convince the Reviewer.”

We do apologize for any technical error that may have occurred during the submission procedure. We are now re-attaching these figures to the end of our Response letter for the Reviewer’s information. 

ELISA analysis revealed no detectable levels of IL-6, IL-10, IL-12 or TNF-alpha in cultured BMMs without LPS (Reviewer Figure 1). Furthermore, immunoblotting showed that protein levels did not alter between the groups of BMMs (with or without RAC1) without stimulation with LPS (Reviewer Figure 2). Please notice that these figures will not be included for publication. 

2/ The text should be copyedited by a professional to improve readability and correct mistakes. A few have been corrected but orthers remain. E.g. stimulated by LPS -> stimulated with LPS.

The manuscript has been linguistically reviewed by a native English-speaking scientist and her contribution has been acknowledged in the manuscript.

 

Reviewer #2:

1. The authors state that “it is known that RAC1-deficiency increases extension of filopodial formation; however, filopodial formation was reduced”. However, on the picture show it seems that filopodia are increased in the KO. Does this mean that the picture presented in Fig. 2C is not representative?. Moreover, the text added in the discussion does not clarify the observation to me.

It is known that RAC1-deficiency reduces filopodial formation in macrophages, but filopodial structures that are formed display increased extension (Wells CM, et al. J Cell Sci. 2004;117:1259–1268). In our study, we have not repeated these experiments; however, images represented in Figure 2C support that macrophages deficient for RAC1 display increased extension of filopodial formation. Please note that filopodial structures are longer, but fewer in number in RAC1-deficient macrophages compared to RAC1-expressing macrophages. This has been clearly stated in the Discussion (page 13, lines 13–16).

2. The authors state that they hopefully will perform more detailed analysis of plaque parameters in the future. However, given that they have paraffin embedded slides and HE stainings, I see no reason why it would not be possible to do a scoring of necrotic core on these sections and one additional Sirius red staining (which is relatively simple to perform).

We have now sectioned the remaining distal parts of aortic arches and stained them with Sirius red, as suggested. The number of red color pixels representing collagen composition, regardless of the extent of intimal thickening, have been quantified as percentages with an image analysis software. There was no difference in the percentage of collagen composition between the aortic sections from mice expressing or lacking RAC1 in macrophages. These data have been mentioned in the Results and presented as supplemental data (S3 Fig). 

In Sirius red-stained aortic sections, we identified necrotic cores that we quantified as the percentage of intimal thickened areas, as suggested. In aortic sections that are lacking RAC1 in macrophages, the size of necrotic cores was significantly reduced. This data has been included as a new figure panel (Figure 3C). One should remember that the extent of intimal thickening is different between the groups (Figure 3A), but we presented data in percentages regardless of the magnitude of intimal thickening.

3. It remains unclear whether IL10 and IL12 were not measured in vivo or not presented. I would also argue to add the measurements of TNF in vivo to the manuscript and to discuss these.

As previously mentioned, we tried to measure in vivo serum levels of IL-10 and IL-12 but they were under detectable levels. These data have been mentioned in the Results (page 11, lines 20–21).

As suggested, we have included the data on the in vivo level of TNF-alpha between the experimental groups (new Figure 4B), although these levels did not reach statistical significance. Data presenting the in vitro levels of IL-10 and IL-12 between the cultured macrophages with or without RAC1 have been removed from Figure 4, as they did not yield any statistical differences. Nevertheless, these data have been mentioned in the Results (page 11, lines 18–21).

---

## [Decision Letter · Decision Letter 2]

3 Sep 2020

Lack of RAC1 in macrophages protects against atherosclerosis

PONE-D-20-06077R2

Dear Dr. Akyürek,

We’re pleased to inform you that your manuscript has been judged scientifically suitable for publication and will be formally accepted for publication once it meets all outstanding technical requirements.

Kind regards,

Michael Bader

Academic Editor

PLOS ONE

Additional Editor Comments (optional):

Reviewers' comments:

Reviewer's Responses to Questions

**Comments to the Author**

1. If the authors have adequately addressed your comments raised in a previous round of review and you feel that this manuscript is now acceptable for publication, you may indicate that here to bypass the “Comments to the Author” section, enter your conflict of interest statement in the “Confidential to Editor” section, and submit your "Accept" recommendation.

Reviewer #1: All comments have been addressed

Reviewer #2: All comments have been addressed

2. Is the manuscript technically sound, and do the data support the conclusions?

Reviewer #1: Yes

Reviewer #2: Yes

3. Has the statistical analysis been performed appropriately and rigorously? 

Reviewer #1: Yes

Reviewer #2: Yes

4. Have the authors made all data underlying the findings in their manuscript fully available?

Reviewer #1: Yes

Reviewer #2: Yes

5. Is the manuscript presented in an intelligible fashion and written in standard English?

Reviewer #1: Yes

Reviewer #2: Yes

6. Review Comments to the Author

Reviewer #1: The authors addressed all points requested by me and the other reviewer and can be published in the current form

Reviewer #2: All of my remarks were addressed adequately by the authors

7. PLOS authors have the option to publish the peer review history of their article (what does this mean?). If published, this will include your full peer review and any attached files.

Reviewer #1: No

Reviewer #2: No

---

## [Editor Report · Acceptance letter]

8 Sep 2020

PONE-D-20-06077R2 

Lack of RAC1 in macrophages protects against atherosclerosis 

Dear Dr. Akyürek:

I'm pleased to inform you that your manuscript has been deemed suitable for publication in PLOS ONE. Congratulations! Your manuscript is now with our production department. 

Kind regards, 

on behalf of

Prof. Michael Bader 

Academic Editor

PLOS ONE